# Function of Retinoic Acid in Development of Male and Female Gametes

**DOI:** 10.3390/nu14061293

**Published:** 2022-03-18

**Authors:** M. Christine Schleif, Shelby L. Havel, Michael D. Griswold

**Affiliations:** School of Molecular Biosciences, Center for Reproductive Biology, Washington State University, Pullman, WA 99163, USA; mary.schleif@wsu.edu (M.C.S.); shelby.havel@wsu.edu (S.L.H.)

**Keywords:** retinoic acid, spermatogenesis, gamete development, meiosis, STRA8

## Abstract

Retinoic acid, an active metabolite of vitamin A, is necessary for many developmental processes in mammals. Much of the field of reproduction has looked toward retinoic acid as a key transcriptional regulator and catalyst of differentiation events. This review focuses on the effects of retinoic acid on male and female gamete formation and regulation. Within spermatogenesis, it has been well established that retinoic acid is necessary for the proper formation of the blood–testis barrier, spermatogonial differentiation, spermiation, and assisting in meiotic completion. While many of the roles of retinoic acid in male spermatogenesis are known, investigations into female oogenesis have provided differing results.

## 1. Introduction

Humans require retinoic acid (RA) for many developmental processes. We can synthesize RA through the metabolic breakdown of retinol. RA is needed for signaling and transcriptional regulation for processes involved in ocular, cardiac, pancreatic, and testicular development [1,2,3]. RA can influence transcription through binding with receptor proteins to help regulate the transcription of certain genes important for these developmental processes in both sexes [4].

The process of producing male gametes is known as spermatogenesis, and if it is disrupted, it results in issues contributing to infertility. Some of the first studies to reveal the importance of RA in spermatogenesis involved feeding rodents a vitamin A-deficient (VAD) diet, resulting in infertile animals incapable of producing viable sperm [5,6]. However, if these VAD rodents were administered exogenous RA, spermatogenesis would be rescued [5,7,8]. Currently, we know RA to be paramount for many functions of spermatogenesis, such as spermatogonial differentiation, the formation of the blood–testis barrier, and the release of mature spermatids into the lumen [9,10,11].

RA has been reported to be involved in female reproduction ranging from meiotic commitment in utero to allowing successful embryo implantation. While a large body of evidence suggests a functional role of RA, evidence concerning whether RA is, in fact, necessary for female germ cell development or for the initiation of meiosis remains controversial [11,12,13,14,15]. RA plays a crucial role in developmental and reproductive processes for both sexes, and its absence can result in severe consequences. This review focuses on the effects of RA on reproductive processes in humans and rodent models, with particular emphasis on male and female gamete formation.

## 2. Male Reproduction

### 2.1. Retinoic Acid in Murine Spermatogenesis

In the study of spermatogenesis, RA is commonly the focal point of study. Past research has looked at testis-specific synthesis of RA. RA is synthesized in a two-step oxidative process carried out by retinol dehydrogenase 10 (RDH10) and the aldehyde dehydrogenase 1A (ALDH1A) family of enzymes (ALDH1A1, ALDH1A2, and ALDH1A3) [4,16,17]. If *Rdh10* is ablated in both Sertoli and germ cells, the resulting testes will be void of RA; thus, spermatogenesis will halt before spermatogonial differentiation [17]. Similar studies knocked out all three *Aldh1a* genes in Sertoli cells, resulting in A spermatogonial cells failing to differentiate into A1 spermatogonia and to continue spermatogenesis [18]. While these catalytic enzymes are vital to RA synthesis in the testis, the timing of RA synthesis is possibly equally as important.

In murine spermatogenesis, a large surge of RA has been observed early in spermatogenesis that coincides with the progenitor cells, also known as A undifferentiated (A_undiff_) spermatogonia, differentiating into A1 spermatogonia [19,20]. Without this pulse of RA, vital genes would not become activated or transcribed, resulting in mitotic arrest of spermatogonia and, ultimately, infertility [21,22]. One such gene, stimulated by retinoic acid gene 8 (*Stra8*), is an RA-activated gene and was previously only known to have effects on meiotic completion [23]. Studies have since identified this RA-Stra8 activation to be important for both spermatogonial differentiation [24]. This study found impaired differentiation and the accumulation of A_undiff_ spermatogonia when *Stra8* was ablated. However, these Stra8-null animals did show that a fraction of their A_undiff_ spermatogonia were capable of differentiating, which the authors suggest is likely a result of RA-mediated activation of other regulatory genes. A subsequent study reaffirms these findings that RA is needed for differentiation, while showing that meiotic initiation and STRA8 expression persist in the absence of RA [16].

Further studies investigated Stra8-null mice and found that pre-meiotic DNA replication was present and meiotic transcripts were expressed, thus concluding Stra8 must be dispensable to meiotic initiation [25]. A recent study by Gewiss et al. confirmed this idea of Stra8 as a regulator of pre-meiotic action. This study used RNA sequencing (RNA-seq) to compare transcripts from isolated germ cells of wild-type and Stra8-null mice prior to meiosis. The results showed transcriptional changes at the time of spermatogonial differentiation, which likely accounted for the previously established meiotic failure in Stra8-null mice [26]. Another study reiterated these findings, using RNA-seq with an adapted culture technique, showing *Stra8* functions in two roles: (1) as a transcriptional repressor prior to differentiation to downregulate pluripotency genes and (2) as a transcriptional activator to activate genes associated with pre-meiotic initiation [27]. These studies make the argument that RA-mediated activation of *Stra8* regulates spermatogonia differentiation and pre-meiotic activation.

RA controls spermatogenesis not only through gene regulation during spermatogenesis but also by facilitating the construction of the essential microenvironment supporting spermatogenesis. The blood–testis barrier (BTB) is a network of tight junctions formed between neighboring Sertoli cells to create an indispensable microenvironment capable of supporting developing germ cells and helping maintain homeostasis. This microenvironment protects the developing cells in seminiferous tubules from interfering molecules in the greater testis. Studies have demonstrated the importance of RA in the formation of the BTB by inhibiting retinoic acid receptor (RAR) signaling in Sertoli cells [28]. RARs are nuclear receptors known to heterodimerize with retinoid X receptors (RXRs) at the promoter regions of genes. Depending on RA interaction, RAR/RXRs can recruit either nuclear repressors or activators to influence transcription [4,29]. Using mice, which overexpressed dominant-negative retinoic acid receptor alpha (dn*Rara*), researchers found that Sertoli cells are unable to complete RA-RAR*a* signaling, consequently revealing stage-specific abnormalities in the development of BTB [28]. Therefore, without RA-RAR signaling, the BTB would not form properly, resulting in an imbalanced microenvironment and subsequent abnormalities in spermatogenesis. Another result of impaired RA signaling is a defect in spermiation. Spermiation is the release of mature spermatids into the lumen of the seminiferous epithelium. When RXRβ is ablated in Sertoli cells, the resulting mice are unable to complete RA-RARα/RXRβ signaling and, thus, are unable to release their spermatids from Sertoli cells [30].

Together, these data reaffirm the notion that RA is mandatory for proper spermatogenesis. RA is needed many times through spermatogenesis (Figure 1). RA first mediates genes, such as Stra8, that are required for spermatogonial differentiation as well as meiotic preparation and completion. Following this, RA facilitates RA-RAR/RXR signaling needed for the regulation of transcripts required to form the tight junctions creating the BTB. Finally, RA aids in the preparation and signaled release of mature spermatids from Sertoli cells.

### 2.2. Spermatogenesis in Humans

The dynamics of human spermatogenesis are similar to that of the murine model, as both include spermatogonia, spermatocytes, and spermatids. These processes sharply diverge, however, when we begin to examine early human spermatogonial development.

In men, there have been three subtypes of spermatogonia identified: dark type A spermatogonia (Ad), pale type A spermatogonia (Ap), and B spermatogonia [31]. Ad and Ap spermatogonia are comparable to A undifferentiated spermatogonia in rodents, but their function is still highly debated. The previously accepted model of human spermatogonial development is that Ad cells are reserved stem cells in a quiescent state, while Ap are in an active proliferative state [32,33]. The newly accepted model asserts that Ap and Ad are more fluid steps of development regulated by transcription of GFRA1 and UTF1. This new study found that, at the start of spermatogenesis, most cells are capable of slow proliferation when GFRA1 is highly expressed. Then, a subset of these cells begin to express UTF1 and reduce the expression of GFRA1, resulting in a quiescent state [34].

By using single-cell RNA sequencing, several stage-specific human germ cell markers (HMGA1, PIWIL4, TEX29, SCML1, and CCDC112) were identified, along with the discovery of differential expression patterns in somatic cells between healthy patients and patients with nonobstructive azoospermia [35]. To elucidate the involvement of these somatic cells within human spermatogenesis and RA action, a recent study used quantitative polymerase chain reaction (q-PCR) to measure the expression of all RAR and RXR isoforms in healthy human testes compared with testes diagnosed with Sertoli cell-only syndrome (SCOS) and maturational arrest (MA) [36]. The resulting data showed severely decreased levels of RAR (α, γ) and RXR (α, β, γ) transcripts in men with SCOS and MA. This study connected these RAR/RXR deficiencies with decreased RA function and suggested that these genes may serve as potential targets for the treatment of SCOS and MA [36]. A recent clinical trial studied men experiencing oligoasthenozoospermia, resulting in infertility issues and provided them with supplemental RA in the form of isotretinoin. This study observed improvements in sperm counts and several successful pregnancies [37]. Together, these studies cultivate promising implications of potential treatments and markers of infertility in men.

### 2.3. Retinoic Acid Inhibition Application as a Potential Contraceptive

The current forms of contraception for men are limited to condoms or vasectomy. However, RA poses many potential avenues for non-hormonal contraception via manipulation of the synthesis or function of RA in the testis. In many cases, the disruption of RA synthesis causes spermatogonia to be halted in their A spermatogonia undifferentiated state [19,22]. One example of a chemical used to disrupt RA synthesis is bis-(dichloroacetyl)-diamine (BDAD). The administration of BDAD inhibits the function of aldehyde dehydrogenases, effectively resulting in the failure to convert retinal to RA [38]. Studies have successfully implemented this idea using BDAD WIN 18,446 on rodents, resulting in arrest of undifferentiated A spermatogonia [39,40]. When given to men, they produced a disulfiram reaction in response to alcohol consumption, which interferes with alcohol metabolic breakdown, making this contraceptive method, although effective, extremely dangerous [41,42]. A more promising approach to a contraceptive utilizing RA manipulation is that of small inhibitory molecules. Many studies have ablated vital RA-dependent genes (*Rara* and *Stra8*), all resulting in infertility and providing potential targets for inhibition [9,43]. A recent study using pan-retinoic acid receptors (pan-RAR) antagonist, BMS-189453, found that, by inhibiting RAR functions, the testis displays an induced sterility, which can be rescued by discontinuing treatment [9,44,45]. In summary, RA acts on many aspects of spermatogenesis, from differentiation of spermatogonia to the formation of the blood–testis barrier. Using this knowledge, we may one day be able to manipulate RA synthesis or signaling pathways to develop safe methods for male contraceptives.

## 3. Female Reproduction

Male and female gamete formation share evidence that suggests a functional role of RA driving reproductive processes critical to germ-cell development. While there is much evidence to support the role of RA within the formation of female gametes, some studies have concluded that the initiation of meiosis is not tied to the action of RA but instead is reliant on other molecular factors present during embryonic development and sexual differentiation.

### 3.1. The Case for Retinoic Acid in the Initiation of Meiosis

As sexual differentiation and meiosis are initiated within the mammalian fetal ovary, the gene *Stra8* has been found to be highly expressed and prompt early meiotic entry necessary for female embryonic germ cells [46]. This largely accepted model asserts that exposure to RA stimulates Stra8, allowing female gametes to enter the first stages of meiosis. While female meiotic initiation is thought to be reliant on the presence of RA, in the embryonic male, it was found that the degradation of RA via the P450 enzyme CYP26B1 is, instead, what determines male sexual fate by keeping embryonic germ cells from entering meiosis until male adolescence [47].

Contrasting evidence has emerged, however, to refute the role of RA in the meiotic induction of embryonic female germ cells. Studies showing no involvement in meiotic initiation used transgenic mouse models deficient in enzymes that synthesize RA (Aldh1a1, Aldh1a2, and Aldh1a3). These studies found that the loss of function in these enzymes resulted in no significant changes in Stra8 expression while meiosis was still successfully initiated [12,15]. However, mice with dietary-induced vitamin A deficiency did not express detectable levels of *Stra8* within female ovaries, resulting in significantly fewer oocytes being able to enter meiosis [8].

Bowles et al. investigated the expression levels of RA in the developing fetus using two methods: β-galactosidase reporter mice and gonadal extract quantification. Within the fetal ovary, mice implanted with the β-galactosidase reporter exhibited low levels of RA, while gonadal extracts taken at the same timepoints revealed high expression levels of RA [48]. Bowles et al. address this discrepancy as likely due to technical limitations of the β-galactosidase reporter model. Their study also measured RA expression in the fetal testis at the time of sexual differentiation, the resulting data showed significantly low levels of RA. This adds additional evidence to support the claim that male germ cells are shielded during embryogenesis by the metabolic breakdown of RA, as facilitated by CPY26B1, thus preventing them from initiating meiosis [48]. To provide direct functional relationships between RA and the initiation of meiosis, Bowles et al. showed that female mice treated with antagonist AGN193109 for retinoic acid receptors exhibited an immediate loss of expression of *Stra8*, *Scp3*, and *Dmc1*. These authors also show that, in the presence of this antagonist, there were fewer meiotic chromosomal figures present. These authors conclude from this work that RA is required to promote normal entry into meiosis in the fetal ovary [48].

Further work within the fetal ovary has shown that, in Aldh1a2-deficient mice, fetal germ cells were able to express *Stra8* and enter meiosis normally, even with a loss in RA signaling [12]. A study performed by Vernet et al. showed the ovaries devoid of RARs were still able to produce meiotic-competent germ cells, which yielded healthy pups [49]. Currently, more studies are required to fully understand the potential role of RA within the embryonic mammalian ovary.

### 3.2. Retinoic Acid Functions to Support Female Fertility

The results described above outline the processes of sexual differentiation and the arguments that exist for and against the theory of RA-mediated meiotic initiation. While meiotic initiation is critical, we must also look for translational work surrounding RA in women’s reproductive health. One such study has been performed by examining germ cells isolated from adult women ranging from 6 to 15 weeks post-fertilization in order to study what regulates meiotic initiation [50]. They found during meiotic initiation at 11 weeks post-fertilization, when it is known that *Stra8* begins to be expressed, the expression of *Aldh1a1* was detected via PCR, leading researchers to conclude that RA is responsible for meiotic entry within female human germ cells [50,51]. This upregulation of Stra8 and other retinoic acid response elements has been broadly classified within human fetal development in females [52,53,54,55].

Outside of critical meiotic induction, there are many studies that have been devoted to determining other effects of RA critical to female reproductive health. One study on human follicular fluid investigated the role of RA in aiding successful fertilization during in vitro fertilization (IVF) fertility treatment. This study found that embryos containing higher levels of RA resulted in better quality and higher fertilization rates [13]. Additionally, the influence of RA has been evaluated in a study looking at granulosa cells isolated from adult patients undergoing IVF. A direct correlation was observed between patients with higher fertilization rates and higher amounts of RA present within their granulosa cells [14].

Overall, this evidence supports multiple roles of RA in maintaining the health of the human female reproductive system, and further innovations into the molecular mechanisms underlying this will continue to pave the way for advancements in healthcare relating to female reproductive health and fertility.

## 4. Conclusions

Within this review, we encapsulated the current understanding of the role of retinoic acid (RA) within the processes that maintain male and female germ cell development. RA has been found to be crucial in male rodents in forming the blood–testis barrier and allowing spermatogonial differentiation, spermiation, and pre-meiotic preparation and completion. The knowledge gained from investigations into both murine and human models may allow for further innovation and discovery, resulting in new possible treatments for infertility and male contraceptives.

The requirement of RA for female germ cell development remains unclear, particularly regarding the role of RA in meiotic initiation. Opposing results for and against RA are made. Outside of this meiotic induction point during embryogenesis, it has been documented and expanded upon that RA has functions critical to fertilization and embryonic implantation. Further work may expand on these initial findings to investigate the mechanisms that underlie these processes leading to potential advancements in treating infertility.

## Figures and Tables

**Figure 1 nutrients-14-01293-f001:**
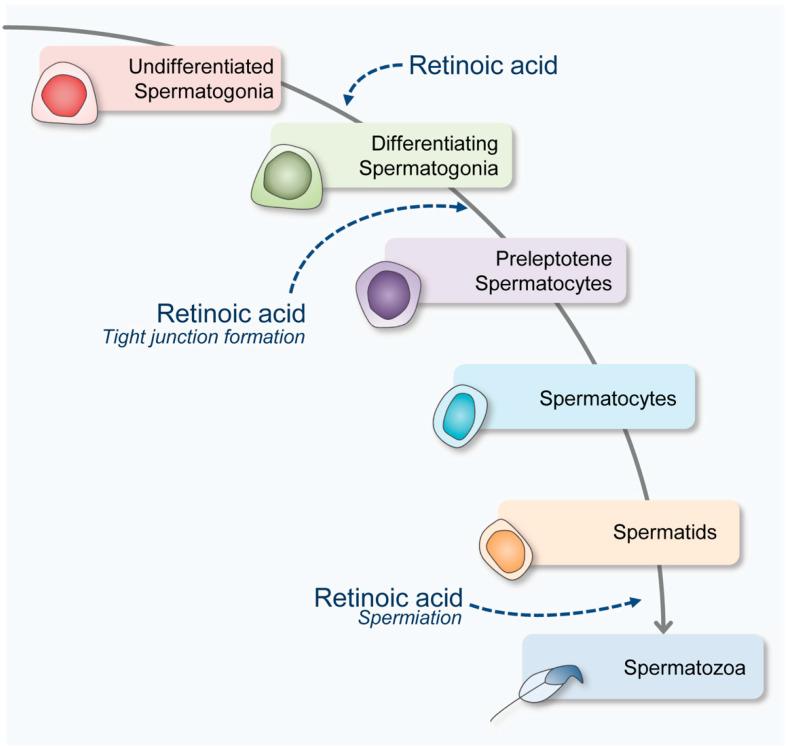
The role of retinoic acid (RA) throughout spermatogenesis. RA is involved in mediating differentiation and pre-meiotic activation in spermatogonia. RA is also needed for signaling within the Sertoli cell to remodel tight junctions associated with the blood–testis barrier. This crucial RA-mediated signaling allows for preleptotene spermatocyte to progress toward the lumen to await their signaled release (spermiation). Spermiation is signaled by RA within the Sertoli cells prior to releasing mature spermatids.

## Data Availability

Data are contained within the article.

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
