# Peer review of "Function of Retinoic Acid in Development of Male and Female Gametes"

_nutrients, 2022, doi:10.3390/nu14061293_

Round 1

Reviewer 1 Report

The Authors present an exhaustive review on the role played by retinoic acid in male and female gametogenesis, also highlighting the controversial aspects that emerge from the literature in studies on this topic.

Minor points:

At lines 66-67: "Further studies investigated Stra8-null mice and found pre-meiotic DNA replication and meiotic transcripts were expressed" should be changed in: "Further studies investigated Stra8-null mice and found that pre-meiotic DNA replication was present and meiotic transcripts were expressed".

I noticed that the reference reported in lines 189-190 (Bowles et al., 2006) is not numbered and is not present in the reference list.

Author Response

Response to Reviewer 1 Comments

Point 1: At lines 66-67: "Further studies investigated Stra8-null mice and found pre-meiotic DNA replication and meiotic transcripts were expressed" should be changed in: "Further studies investigated Stra8-null mice and found that pre-meiotic DNA replication was present and meiotic transcripts were expressed".

Response 1: The authors agree with these edits and have applied them to the updated manuscript.

Point 2: I noticed that the reference reported in lines 189-190 (Bowles et al., 2006) is not numbered and is not present in the reference list.

Response 2: The authors have fixed this error and updated the citations and references.

Reviewer 2 Report

This review by Pr M. D. Griswold et al. is extremely well written. It summarizes the current knowledge on the role of retinoic acid (RA) in germ cell differentiation and in fertility. Efforts were made to integrate most, if not all, the studies published up to now, including some refuting the role of RA in meiotic initiation. Nonetheless, a review should concentrate on the facts, staying as neutral as possible, without infusing bias towards an opinion or another. The reviewer detected some sentences which indicate a preference for the authors to the idea that RA “is” instructing meiosis. For instance, the notion that meiosis initiates despite the lack of RA in mutant mice lacking all RA-synthesizing enzymes in their seminiferous epithelium is skipped. Along the same lines, the fact that germ cells initiate meiosis in ovaries devoid of all three RAR isotypes is also skipped. In addition, some references in the text do not exactly referred to what the authors wrote.

Major points

  • Lines 13, 33, 166 and 207: “as well as meiotic initiation”, “ranging from meiotic commitment”, ”evidence to support the role of RA” and “RA-mediated meiotic initiation” are examples of the bias toward the idea that RA “is” the meiotic inducer.
  • Lines 60-61: the authors wrote that “Studies have since identified RA-Stra8 activation to be important for […] and meiotic completion [24]”. To be complete and fully impartial, they should also cite another study showing that meiotic initiates normally despite the absence of RA in the seminiferous epithelium (ref 16).
  • Lines 188-192: the authors cite the work published by Bowles et al. (2006) showing that RA-dependent activity exists in the foetal ovary, while not in the foetal testis at the same development stage. With all respect, the study primarily shows that RA activity is strong in the mesonephros (but very low in the ovary). Here, to be impartial, the authors should cite the study by Kumar et al. (2011) [ref 12], which shows that RA-dependent activity is totally lost in ALDH1A2-null mutants while their germ cells are expressing Stra8 and entering meiosis normally.
  • Lines 192-195: the authors wrote that “mice treated with both agonists and antagonists for RA receptors [show] that RA directly stimulates Stra8 expression and entry to meiosis” and “initiation of meiosis can be induced by activating RA-receptors”. Here again, to be impartial, they should cite the recent study by Vernet et al. (2021), which show that ovaries devoid of RARs produce meiotic germ cells that are capable to give rise to live pups [PMID:32917583]. In addition, the authors should consider the fact that the retinoids used in most, if not all, the cited studies (refs 46, 47) were not neutral RAR antagonists, but inverse agonists, the use of which do not necessarily reflect the natural functioning of RA-activated RARs [PMID:19477412]. By the way, ref 48 cited line 195 relates to the foetal testis not to the ovary, while ref 49 deals with the effect of RA added in cultured ovaries on the apoptosis of germ cells. None of them shows that "RA directly stimulates Stra8 expression and entry to meiosis".
  • Lines 197-198: the authors state that “evidence exists to support this relationship, especially when we compare the activity of RA in the ovary with how it acts in the testis”. Why the mechanisms controlling germ cell fate should identical in the ovary and in the testis? Does not this sentence reflect the authors' preconceived notions about the action of RA on germ cells?

Minor points

  • Line 89: ref 29 does not relate to interaction of RAR/RXR with coactivators or corepressors, but to the morphological defects displayed by RAR knockout mutants and found in vitamin A deficiency. The authors may cite Perissi & Rosenfeld (2005) [PMID:15957004].
  • Line 146: ref 8 does not relate to the effect of RA on spermatogonia, but of vitamin A deficiency. The authors may cite refs 16-18.
  • Line 155: Rarb is mentioned as an example of vital RA-dependent gene, the ablation of which results in infertility. However, Rarb knockout mutants are fertile [PMID: 8555112 and PMID:9240560]. The authors should cite refs 1 and 9.
